# Near-atomic structure of a giant virus

Qianglin Fang[1], Dongjie Zhu [2,3], Irina Agarkova[4], Jagat Adhikari[5], Thomas Klose [1], Yue Liu [1], Zhenguo Chen[1], Yingyuan Sun[1], Michael L. Gross [5], James L. Van Etten[4], Xinzheng Zhang [2,6] & Michael G. Rossmann[1]

Although the nucleocytoplasmic large DNA viruses (NCLDVs) are one of the largest group of viruses that infect many eukaryotic hosts, the near-atomic resolution structures of these viruses have remained unknown. Here we describe a 3.5 Å resolution icosahedrally averaged capsid structure of Paramecium bursaria chlorella virus 1 (PBCV-1). This structure consists of 5040 copies of the major capsid protein, 60 copies of the penton protein and 1800 minor capsid proteins of which there are 13 different types. The minor capsid proteins form a hexagonal network below the outer capsid shell, stabilizing the capsid by binding neighboring capsomers together. The size of the viral capsid is determined by a tape-measure, minor capsid protein of which there are 60 copies in the virion. Homologs of the tape-measure protein and some of the other minor capsid proteins exist in other NCLDVs. Thus, a similar capsid assembly pathway might be used by other NCLDVs.

[1] Department of Biological Sciences, Purdue University, West Lafayette, IN 47907, USA. [2] National Laboratory of Biomacromolecules, CAS Center for Excellence in Biomacromolecules, Institute of Biophysics, Chinese Academy of Sciences, 100101 Beijing, China. [3] School of Life Science, University of Science and Technology of China, Hefei 230026, China. [4] Department of Plant Pathology and Nebraska Center for Virology, University of Nebraska-Lincoln, Lincoln, NE 68583-0900, USA. [5] Department of Chemistry, Box 1134, Washington University, One Brookings Drive, St. Louis, MO 63130, USA. [6] University of Chinese, Academy of Sciences, Beijing 100049, China. These authors contributed equally: Qianglin Fang, Dongjie Zhu. Correspondence and requests for materials should be addressed to X.Z. (email: xzzhang@ibp.ac.cn) or to M.G.R. (email: mr@purdue.edu)

The nucleocytoplasmic large DNA viruses (NCLDVs), including Phycodnaviruses, Mimiviruses, Iridoviruses, Asfarviruses, Ascoviruses, Marseilleviruses, Pandoraviruses, and Poxviruses, are one of the largest group of viruses that infect a broad variety of eukaryotic hosts. These viruses have genomes[1] of between 100 kb and 2.5 Mb with some shared genetic traits[2,3]. Except for Poxviruses, Pandoraviruses, and Ascoviruses, all these viruses are roughly icosahedral in shape.

Previously reported cryo-EM reconstructions of these approximately icosahedral NCLDVs are Paramecium bursaria chlorella virus 1 (PBCV-1)[4–6], Chilo irridescent virus (CIV)[4,7], Phaeocystis pouchetti virus (PpV01)[8], Melbourne virus[9], Cafeteria roenbergensis virus (CroV)[10], Acanthamoeba polyphaga mimivirus (APMV)[11], faustovirus[12], and pacmanvirus[13]. These cryo-EM reconstructions showed that these viruses have some common structural features. Most of these viruses have an icosahedral or roughly icosahedral external capsid assembled from hundreds of pseudo-hexagonal, close-packed, trimeric capsomers and an inner membrane that separates the nucleocapsid from the external capsid. The capsomers of the external capsid are organized into 20 threefold-axes-centered triangular arrays and 12 fivefold-axes-centered pentagonal arrays named trisymmetrons and pentasymmetrons, respectively. The pentasymmetrons consistently contain 31 capsomers (30 trimers and a pentamer). In addition, minor capsid proteins, internal to the major capsid proteins and outside the lipid membrane can be recognized in those cryo-EM reconstructions that have better than ~15 Å resolution[6,7,12]. The common structural features of these approximately icosahedral NCLDVs indicate that these viruses probably use similar principles in assembling their capsids.

A previously reported 10 Å resolution, icosahedrally averaged, cryo-EM reconstruction of PBCV-1 capsid is currently the highest-resolution structure available for any of these approximately icosahedral NCLDVs[6]. The large size (150–500 nm in diameter) and potential flexibility of the viral capsids might have impeded structure determinations to higher resolution[14,15]. Near-atomic-resolution structures are required to establish how these viral capsids are assembled. Given the recent resolution revolution[16] in the use of cryo-EM, the resolution limit of these virus reconstructions can probably be improved. Because PBCV-1 has a relatively small size (~190 nm in diameter) among these NCLDVs, it is a good candidate for exploring this possibility. An increase in resolution might establish the function of the minor capsid proteins and suggest various assembly pathways such as how the size of the icosahedrally symmetric capsid is controlled during assembly.

PBCV-1, a member in the chlorovirus genus of the family *Phycodnaviridae*[17,18], is a large, dsDNA-containing virus that infects certain chlorella-like green algae. It has a 330-kbp genome that encodes 416 predicted proteins and 11 tRNA molecules[19]. A proteomic study identified 149 different proteins in the mature PBCV-1 virion[19]. The PBCV-1 nucleocapsid is enclosed by a lipid bilayer membrane[4]. The membrane is surrounded by a roughly icosahedral external capsid shell that has a unique vertex where a spike structure[5], required for host entry[6], is located. The external capsid is constructed of capsomers that are arranged into tri- and pentasymmetrons. Each tri- and pentasymmetron contain 66 and 31 capsomers (30 pseudo-hexameric capsomers and a pentameric capsomer), respectively. The atomic structure of the major capsid protein (MCP), Vp54, was determined by X-ray crystallography and consists of two sequential jelly-roll folds with four N-linked glycans[20,21]. Each pseudo-hexameric capsomer consists of three copies of the MCP arranged with pseudo-6-fold symmetry relating the six jelly-roll domains. The overall arrangement of the capsomers in the icosahedral lattice follows $T = 169d$ triangulation[22]. The cleavage line between neighboring symmetrons is caused by a pseudo-6-fold rotation between capsomers on either side of the boundary between neighboring symmetrons[23]. Below the outer capsid shell there are several minor capsid proteins, presumably for stabilizing the viral capsid[6].

Here we have extended the resolution of the PBCV-1 structure to 3.5 Å by collecting data using a direct electron detector. Furthermore, the resolution improvement benefitted greatly by developing a method to account for the defocus gradient in large virus particles and for dealing with local flexibility of the virus particles[24]. The improved resolution of the cryo-EM map resulted in the identification of 13 different minor capsid proteins and an atomic model of the viral capsid consisting of 6900 polypeptide chains. This is by far the largest viral capsid structure that has been determined to near-atomic resolution, and the first near-atomic description of a NCLDV capsid. The structure suggests that the PBCV-1 capsid is assembled via a pathway that is likely to be similar in other approximately icosahedral NCLDVs.

## Results

**Overall structure of PBCV-1.** A 4.4 Å resolution, icosahedrally averaged reconstruction of PBCV-1 was calculated using conventional image processing programs[25] by averaging ~13,000 particles from 5624 images recorded with a direct electron detector (Supplementary Fig. 1a). This compares with the earlier 10 Å resolution structure obtained by using film data[6]. This resolution was still not sufficient for de novo atomic modeling. However, further improvement of the resolution to 3.5 Å was obtained by correcting for the defocus gradient through large virus particles and by correcting for the flexibility of the viral capsid[24] (Fig. 1a and Supplementary Fig. 1a, b). These significant improvements in resolution of the PBCV-1 cryo-EM map made it possible to model the viral capsid at near-atomic resolution (Supplementary Fig. 2).

The atomic structures of the 28 pseudo-hexameric capsomers in the icosahedral asymmetric unit were obtained by fitting the previously reported atomic model of Vp54 (the MCP) pseudo-hexamers as rigid bodies into the cryo-EM map. Each amino acid residue of each independent Vp54 molecule was then adjusted by hand to improve the fit into the density using the program COOT[26], followed by real-space refinement using the program PHENIX[27].

The main chain of many of the remaining uninterpreted parts of the map representing minor capsid proteins as well as the penton proteins around the 5-fold vertices were built using the EMBuilder program[28] and then rebuilt manually using the program COOT[26]. Because the identities of the minor capsid proteins and the penton protein were unknown, a program was designed to match the side chain size distributions observed in the cryo-EM map with the protein sequences determined by a proteomic study of the mature PBCV-1 virion[19] (Methods). This led to the identification of the penton protein (P1) and 13 different minor capsid proteins (P2, P3, P4, P5, P6, P7, P8, P9, P10, P11, P12, P13, and P14) (Supplementary Table 1). The minor capsid proteins and penton protein were found to be conserved in 40 other chloroviruses[29,30] that infect four different algae, indicating that they presumably have significant roles in the life cycle of chloroviruses.

The outer capsid shell of the icosahedrally averaged PBCV-1 structure consists of the MCP (Vp54) and the penton protein. The MCP and penton protein exist as pseudo-hexameric trimers and pentamers, respectively. The pseudo-hexameric capsomers are organized into 20 triangular trisymmetrons and 12 pentagonal pentasymmetrons (Fig. 1a and Supplementary Fig. 3a). In addition, there is one pentameric capsomer that consists of five penton protein monomers in each pentasymmetron

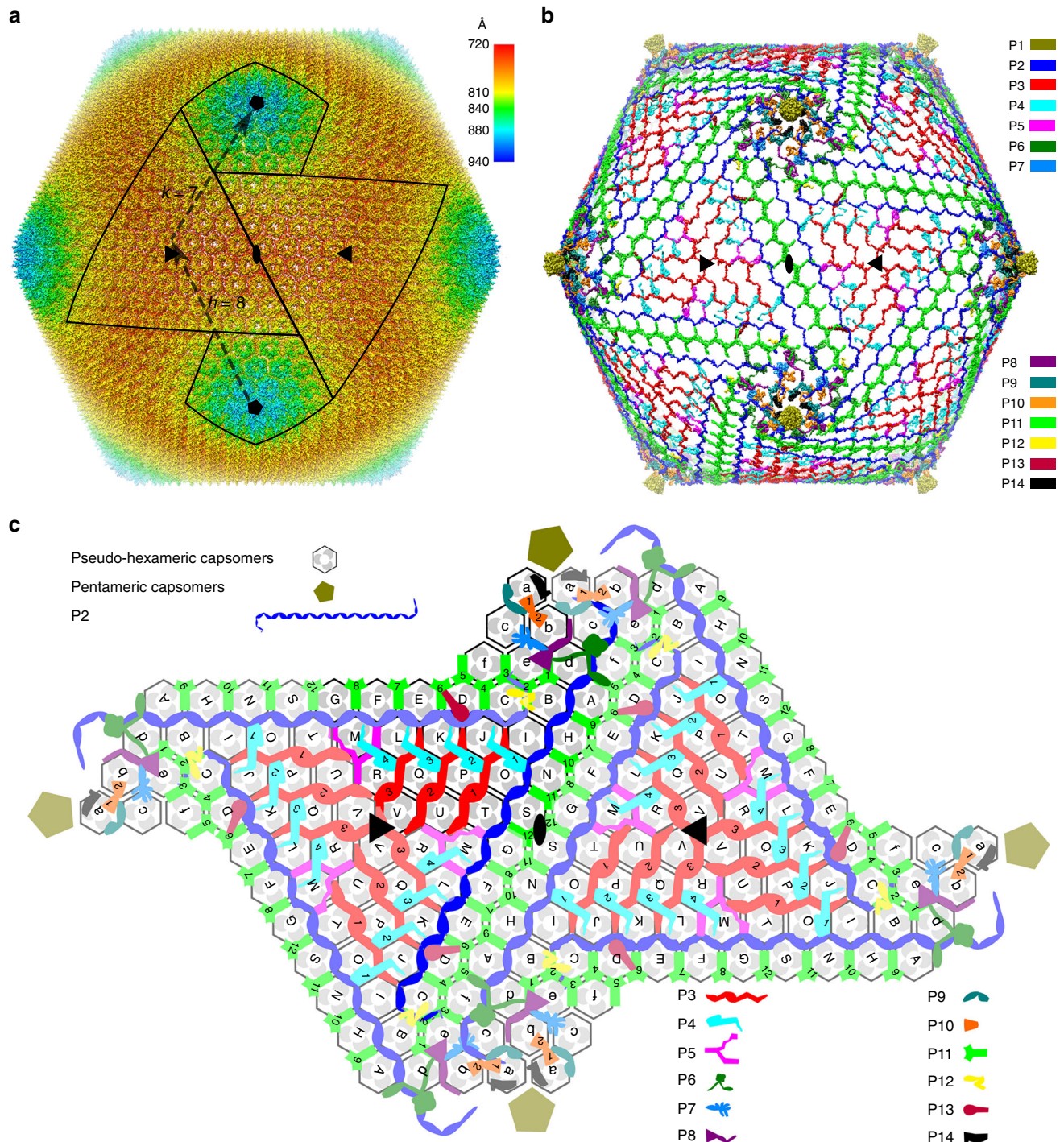

**Fig. 1** Overall structure of the PBCV-1 capsid. **a** Icosahedrally averaged cryo-EM reconstruction of the viral capsid, colored according to the radial distance from the center of the virus. The boundaries of two neighboring trisymmetrons and two pentasymmetrons are outlined in black. The *T* number including the h and k vectors are indicated. **b** The cryo-EM density of the minor capsid proteins and the penton proteins after removing the outer capsid shell. Each protein is shown in a different color as indicated on the right. **c** Diagrammatic organization of the minor capsid proteins and capsomers viewed from inside the virus. The pseudo-hexameric capsomers are outlined. Each gray dot within each hexagon represents a Vp54 subunit. The icosahedral 3-fold and 2-fold axes are shown as solid black triangles and ovals, respectively. Different minor capsid proteins and the penton proteins are shown as different shapes with different colors, as indicated. Darker color is used in one icosahedral asymmetric unit. The pseudo-hexameric capsomers are labeled A, B, C, … in the trisymmetrons and a, b, c, … in the pentasymmetrons

(Supplementary Fig. 3a). Each icosahedral asymmetric unit of the outer capsid shell contains 28 pseudo-hexameric capsomers with 6 of these in the pentasymmetron (a, b, c, d, e, and f) and 22 in the trisymmetron (A, B, C, D, E, F, G, H, I, J, K, L, M, N, O, P, Q, R, S, T, U, and V) (Fig. 1a, c). The minor capsid proteins (P2, P3,

P4, P5, P6, P7, P8, P9, P10, P11, P12, P13, and P14) form a hexagonal network that is immediately below (inside) the outer capsid shell (Fig. 1b, c). Most of these minor capsid proteins are at the interface between neighboring capsomers and probably function to stabilize the whole capsid. P2, P5, P6, P7, P8, P9,

P12, P13, and P14 are present as a single copy in each icosahedral asymmetric unit, whereas the copy number per icosahedral asymmetric unit of P3, P4, P10, and P11 are three, four, two and 12, respectively (Fig. 1b, c).

**The penton protein and the major capsid protein**. The penton protein forms pentameric capsomers positioned at the 5-fold vertices of the outer capsid shell of PBCV-1. All the residues of the penton protein, P1, except for its last two residues, were built into the cryo-EM density map with a good agreement between the amino acid sequence and the density of the cryo-EM map. The penton protein folds into a canonical jelly-roll structure[31] composed of eight ß-strands, arranged into the standard two four-stranded β-sheets (BIDG and CHEF) (Fig. 2a, b). As in other jelly-roll structures, the two four-stranded sheets are stabilized primarily by their hydrophobic interface.

The penton protein has branched electron densities attached to four Asn residues (N21, N93, N129, and N137), indicating the presence of N-linked glycans at these sites. Except for N21, these N-linked glycosylation sites ($AN_{21}TTP$, $GN_{93}VFF$, $GN_{129}VLT$, and $GN_{137}EHS$) are not associated with the canonical NX(T/S) sequon usually recognized by endoplasmic reticulum-located glycosyltransferases. Similarly, the four glycosylation sites on the MCP Vp54 are associated with the non-canonical sequences $AN_{280}IPG$, $GN_{302}TGT$, $GN_{399}TET$, and $AN_{406}TAT$[21]. It is noteworthy that the sites of N-linked glycosylation of the MCP and the penton protein are all preceded by a small residue (Ala or Gly). Thus, the penton protein and the MCP likely share common glycosyltransferases unique to chloroviruses.

The closest structure to the penton protein found in a DALI search[32] is the structure of the penton protein of Sputnik virus[33]. The penton protein subunits of PBCV-1 and Sputnik virus adopt similar orientations with respect to their central 5-fold axes. The primary difference between the penton protein of PBCV-1 and the Sputnik virus are that the PBCV-1 penton protein lacks a large insertion that forms a small domain on top of the penton protein. In addition, there are no glycans attached to the Sputnik virus penton protein.

Each pseudo-hexameric capsomer is composed of three Vp54 (the MCP) subunits. Each icosahedral asymmetric unit of the PBCV-1 structure contains 28 pseudo-hexameric capsomers, giving a total of 84 Vp54 subunits in the icosahedral asymmetric unit of the virus. The structures of these Vp54 subunits are closely similar to that of the previously reported crystal structure of Vp54[21]. The coordinates of chain A of the best resolved Vp54 structure (PDB accession code: 5TIP) were used to compare with the atomic models of these 84 Vp54 subunits. The root-mean-square deviations (rmsd) between the equivalent Cα atoms of the crystal structure and the atomic models of the 84 Vp54 subunits ranged from 0.3 to 1.1 Å. However, bigger differences occurred at the N-terminal 24 residues of a Vp54 subunit of capsomer a and a Vp54 subunit of capsomer b triggered by making contacts with some of their neighboring minor capsid proteins (Fig. 2c). The averaged interface area[34] between any pseudo-hexameric capsomer and its capsomer neighbors, and between any pseudo-hexameric capsomer and its neighboring minor capsid proteins are 5800 Å² and 5600 Å², respectively. Thus, the interface areas between major and minor capsid proteins are about the same as between neighboring capsomers, indicating that the minor capsid proteins play important roles in stabilizing the viral capsid.

**Minor capsid proteins that stabilize trisymmetrons**. The 66 capsomers within each trisymmetron are glued together by a hexagonal network formed by four different minor capsid proteins (P2, P3, P4, and P5) (Fig. 1b, c). Both P2 and P3 have extremely extended, fiber-like conformations, located in the gaps between neighboring capsomers and facing the inside of the virus (Figs. 1c and 3a, b). Minor capsid proteins P2 and P3 have similar, extended polypeptide structures (Fig. 3a, b), but are quite different in their lengths, namely ~720 Å and ~200 Å, respectively. Like P2 and P3, P5 fills the gaps between neighboring

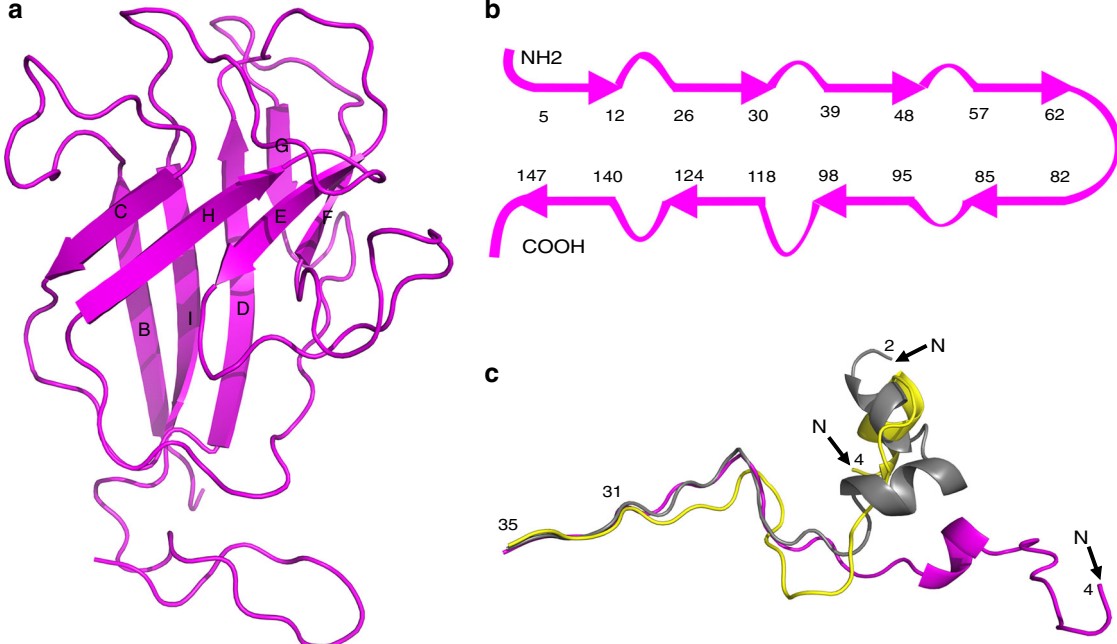

**Fig. 2** Structures of the penton protein (P1) and the MCPs. **a** Ribbon diagram of the penton protein. **b** Diagrammatic representation of the arrangement of β-strands in the penton protein. Residues are numbered at the ends of the β-strands. **c** Different conformational arrangements of the N terminus of the MCP. The Vp54 subunit structure of capsomer a (yellow) and capsomer b (magenta) are superimposed on the crystal structure of Vp54 (gray). Only the N-terminal 35 residues of each are shown for clarity

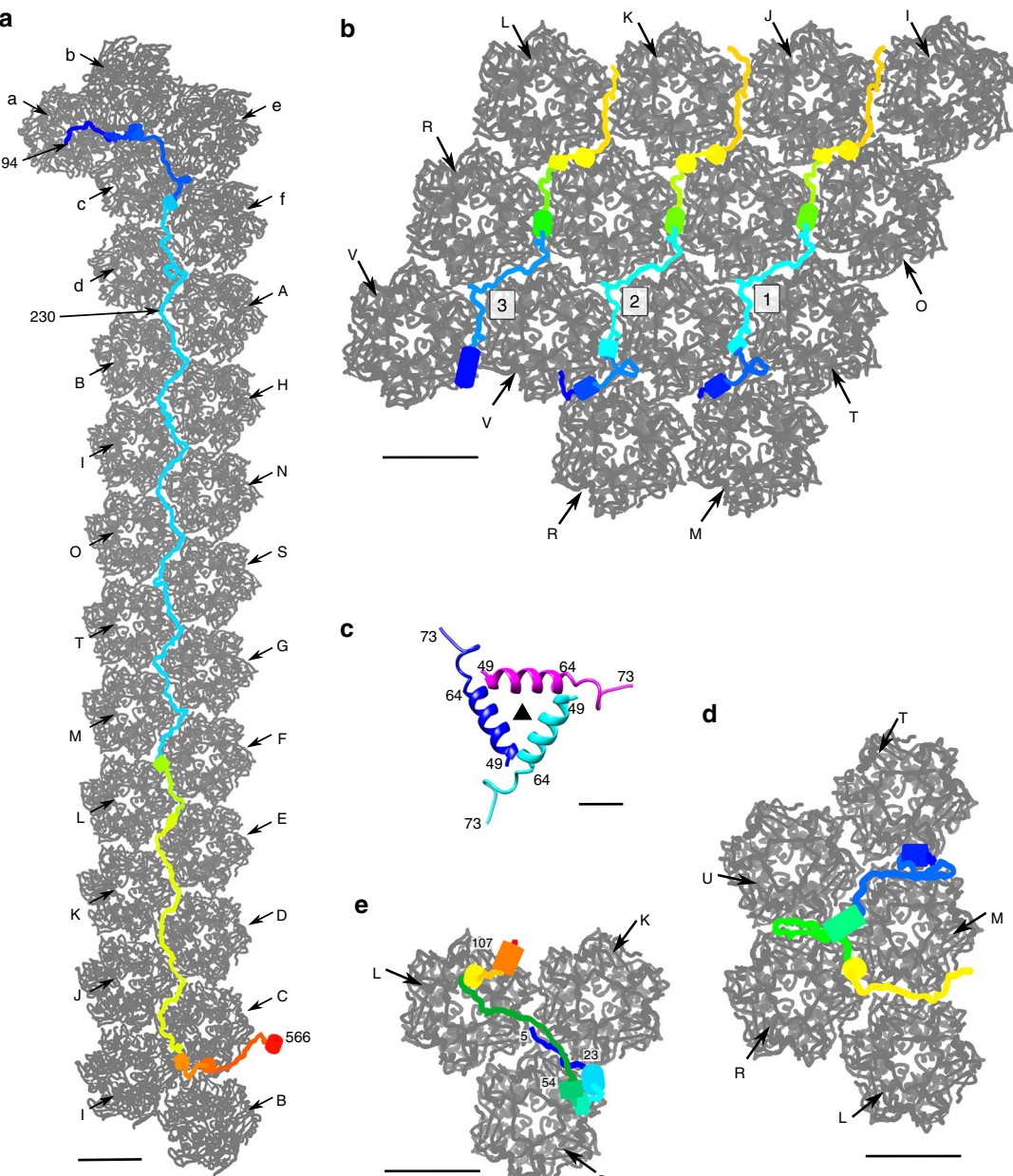

**Fig. 3** Structures of the minor capsid proteins that glue together capsomers within each trisymmetron. Each minor capsid protein is rainbow colored from red at the C terminus to magenta at the N terminus. The neighboring capsomers are shown in gray and labeled as in Fig. 1c. **a** The minor capsid protein P2 (blue in Fig. 1c). Scale bar, 50 Å. **b** The three P3 molecules within one icosahedral asymmetric unit (red in Fig. 1c) and their neighboring capsomers. Scale bar, 50 Å. **c** Ribbon diagram of the three-helix bundle formed by the N-terminal parts of each P3 molecule around each icosahedral 3-fold axis. The icosahedral 3-fold axis is identified by a solid black triangle. Scale bar, 10 Å. **d** The minor capsid protein P5. Scale bar, 50 Å. **e** One of the four P4 molecules within an icosahedral asymmetric unit. Scale bar, 50 Å

capsomers (Figs. 1c and 3d) but folds to a Y-shaped structure (Fig. 3d). The N-terminal part of P4 (residues 5–23) glues together two neighboring capsomers (Fig. 3e). Its central domain (residues 24–54) is folded into a fist-like structure consisting of three short helices (Fig. 3e). The C-terminal part of P4 anchors in the central 3-fold cavity of a third capsomer (Fig. 3e).

Each icosahedral asymmetric unit contains three P3 minor capsid proteins that glue together four neighboring rows of capsomers (Fig. 1c). The structures of the first and second P3 proteins resemble each other with an rmsd of 2.0 Å between equivalent Cα atoms. The amino terminus of the third P3 protein, together with its icosahedral 3-fold related counterparts, forms a three-helix bundle around the icosahedral 3-fold axis of each

trisymmetron (Fig. 3c). The nine copies of P3 that are in each trisymmetron link a set of 36 capsomers, forming a triangular array of capsomeres with eight capsomers per edge centered on the icosahedral 3-fold axis (Fig. 1c). Thus, the three-helix bundle formed by three P3 molecules may be the initiation of the assembly of a trisymmetron. At the completion of the trisymmetron assembly, three copies of P2 bind along the three edges of the triangular capsomer array (Fig. 1c). The amino terminus (residues 94–230) of the three P2 molecules extend to the neighboring pentasymmetrons, and finally stop near the icosahedral 5-fold axis of neighboring pentasymmetrons (Figs. 1b, c and 3a), thus establishing the position of the next 5-fold vertex in the assembly of the virus. These observations suggest that P2,

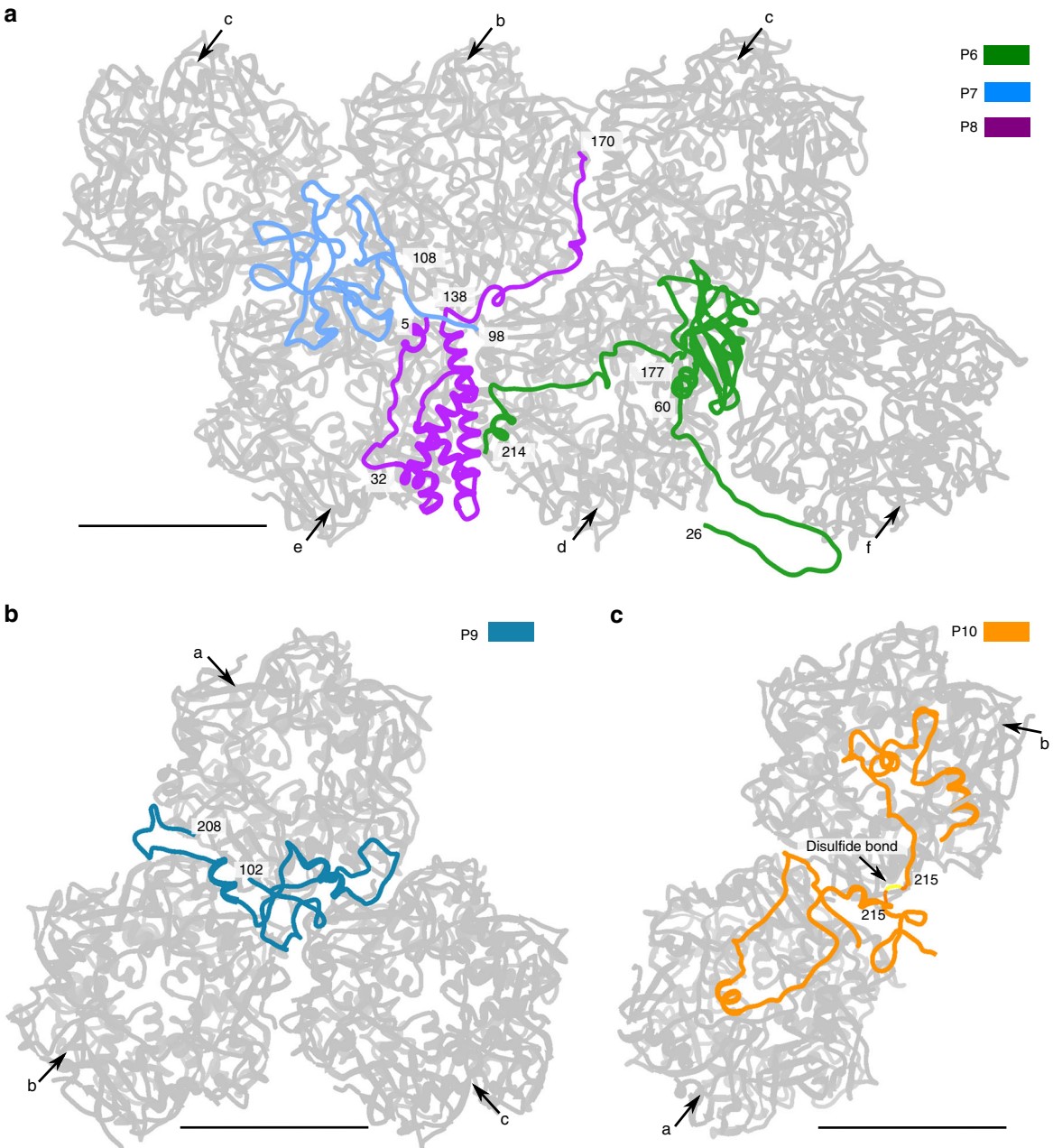

**Fig. 4** Structures of the minor capsid proteins that glue together capsomers within each pentasymmetron. The minor capsid proteins and their neighboring capsomers are shown as ribbon diagrams. All the capsomers are colored gray and labeled as in Fig. 1c. **a** Minor capsid proteins P6, P7, and P8. **b** Minor capsid protein P9. **c** Minor capsid protein P10. Scale bars, 50 Å

together with P3, assemble a scaffold that glues together all the 66 capsomers that occur in each trisymmetron, determining the final size of each trisymmetron and providing a finite limit for each trisymmetron in the assembly of the virus.

**Minor capsid proteins that stabilize pentasymmetrons**. The 30 pseudo-hexameric capsomers in each pentasymmetron are held together by the N-terminal part of P2 (residues 94–230), P6, P7, P8, P9, and P10 (Figs. 1b, c and 3a). Unlike the minor capsid proteins that stabilize the trisymmetrons, the structures of most of the minor capsid proteins in the pentasymmetrons are less extended. P6 has an N-terminal arm (residues 26–60), a central globular domain that consists of a five-stranded antiparallel ß-

sheet wrapped around a central α-helix and flanked by another α-helix, and a C-terminal arm (residues 177–214) (Fig. 4a). P8, like P6, has an N-terminal (residues 5–32) and a C-terminal arm (residues 138–170) (Fig. 4a). The central region (residues 33–137) of P8 folds into a wedge-shaped structure that consists of three short α-helices wrapped around a long α-helix (Fig. 4a). Although P7 consists primarily of loops, it is folded into a globular domain stabilized by three intramolecular disulfide bonds (C120-C134, C156-C172, and C188-C211) (Supplementary Table 2). The V-shaped P9 protein (Fig. 4b) and the P10 protein (Fig. 4c) exhibit slightly more extended conformations than do P6, P7, and P8. Like P7, P9 has two intramolecular disulfide bonds (C104-C131 and C194-C201) that stabilize its structure (Supplementary Table 2).

The N-terminal region of P2 (residues 94–230) holds together capsomers b, c, d, e, and f by gluing the interfaces between capsomers (Figs. 1c and 3a). Furthermore, P2 binds to the internal face of capsomer a, gluing together all six pseudo-hexameric capsomers in the asymmetric unit of a pentasymmetron (Figs. 1c and 3a). P9 bridges the gaps between capsomers a and b, and a and c, gluing together a set of three capsomers (Figs. 1c and 4b).

P6, P7 and P10 are located below P2, suggesting that P2 probably contributes to recruiting these three proteins to the viral capsid (Fig. 1b, c). The central globular domain of P6 is located at the conjunction of capsomers c, d, and f with its C-terminal arm making contacts with the central 3-fold cavity of capsomer d (Figs. 1c and 4a). P7 is situated at the conjunction of capsomers b, c, and e and glues these three capsomers together (Figs. 1c and 4a). There are two copies of P10 in each icosahedral asymmetric unit (Figs. 1c and 4c). The first and second P10 molecules makes contacts with the internal surfaces of capsomer a and capsomer b, respectively (Figs. 1c and 4c). The two P10 molecules are cross-linked by an intermolecular disulfide bond (C215–C215) (Fig. 4c). Thus, capsomer a and capsomer b are bound together by two P10 molecules.

P8 binds together capsomers b, c, d, and e (Figs. 1c and 4a). In addition to binding neighboring capsomers together, P8 also interlocks with P7 at its N terminus and P6 at its C-terminal arm (Figs. 1c and 4a), which all contribute to stabilizing the pentasymmetrons.

**Minor capsid proteins that bind neighboring symmetrons.** There are 12 copies of the zip protein, P11, per icosahedral asymmetric unit (Fig. 1c), the highest copy number among the 13 types of minor capsid proteins. The structures of all the P11 proteins consist primarily of intrinsically flexible loops (Fig. 5a). The P11 proteins zip together most of the capsomers at the boundary of neighboring tri- or pentasymmetrons via their C-terminal regions (residues 168–206) (Fig. 5a). The C-terminal regions of each P11 protein is located underneath the conjunction of a set of three neighboring capsomers (Figs. 1c and 5a), gluing these capsomers together. Two out of the three capsomers that are from the same symmetron have roughly identical orientations, whereas the third capsomer, which is from a different symmetron, rotates by ~60° with respect to the other two capsomers. Such a spatial arrangement of neighboring capsomers occurs only at the symmetron boundaries and at some regions within pentasymmetrons (Fig. 1c).

**Membrane association.** Unlike most of the other minor capsid proteins, three (P12, P13, and P14) (Figs. 1c and 5b) of the 13 identified minor capsid proteins do not serve a cross-linking function between neighboring capsomers, suggesting that they play other roles in the viral capsid assembly. Transmembrane-region predictions using the computer programs TMHMM[35,36], HMMTOP[37,38], and Phobius[39] showed that two regions at the N terminus of P12 (residues 20–42 and 50–67), a region at the

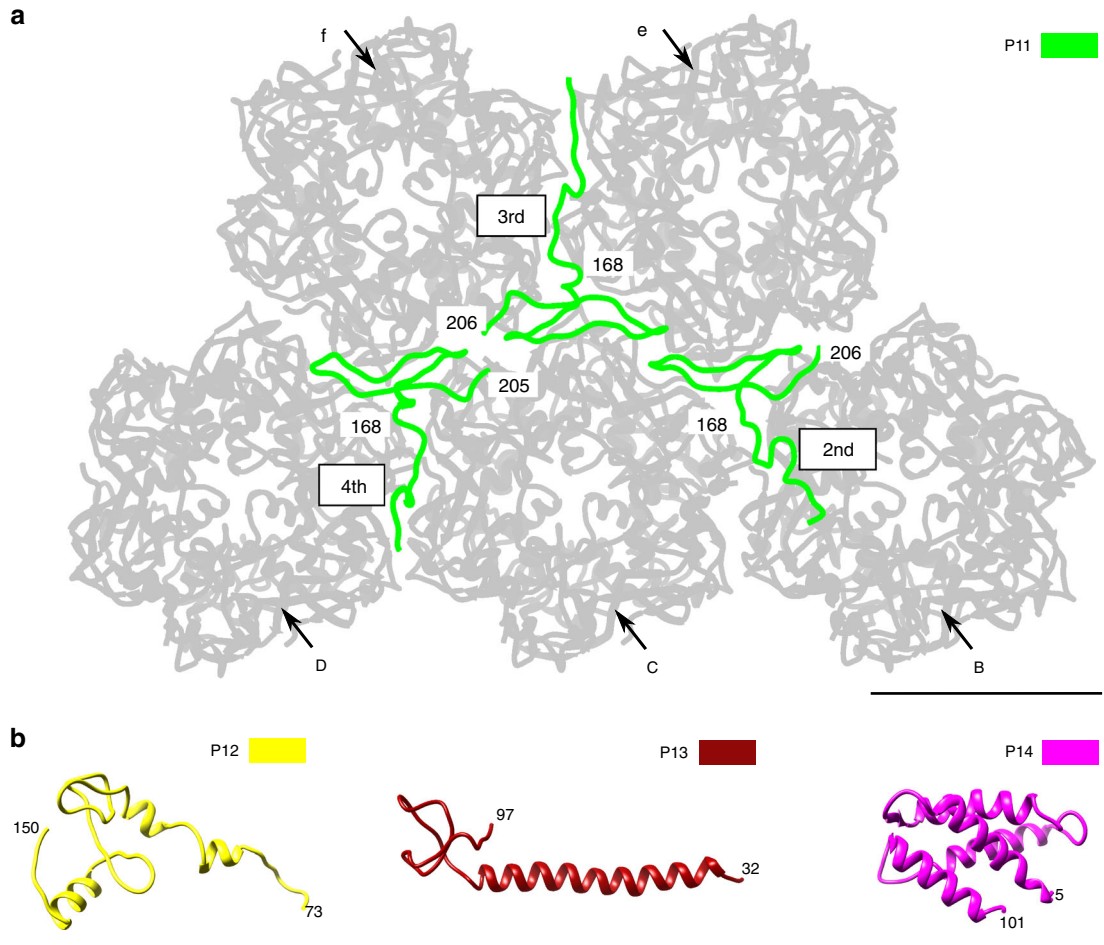

**Fig. 5** Structures of minor capsid proteins P11, P12, P13, and P14. These proteins bind together neighboring symmetrons and/or have potential membrane association functions. All the structures are shown as ribbon diagrams. **a** The second, third, and fourth P11 molecule within an icosahedral asymmetric unit (see Fig. 1c) and their neighboring capsomers (gray). All the capsomers are labeled as in Fig. 1c. **b** Minor capsid proteins P12, P13, and P14. Scale bars, 50 Å

N terminus of P13 (residues 4–23), and a region at the C terminus of P14 (residues 243–265) may all be transmembrane helices (Supplementary Table 3). Further transmembrane-region predictions for the other minor capsid proteins showed that the N terminus of P2, P6, P7, P9, P10, and P11 also have transmembrane helix properties (Supplementary Table 3). Thus, these proteins (P2, P6, P7, P9, P10, P11, P12, P13, and P14) might play roles in associating the viral capsid with the inner viral membrane.

All the predicted transmembrane helices are in disordered regions of the identified minor capsid proteins. Secondary structure predictions[40] showed that the linkers between the predicted transmembrane helices and the ordered portions of each protein consist mainly of flexible loops. Thus, the reason why the transmembrane helices are not visible in the cryo-EM map is that they cross the membrane in slightly different positions in different parts of the virus that are averaged out by icosahedral symmetry. The lengths of the linkers between the visible part of each protein and the predicted transmembrane helices varies from 5 to 171 residues (Supplementary Table 3). The proteins that are closer to the capsid vertices generally have longer linkers, which is consistent with the distance variation between the viral capsid shell and the inner viral membrane. P6, P12, and P13 have very short linkers of less than 9 amino-acid residues (Supplementary Table 3), indicating that the visible portions of these proteins are close to the membrane. The inward-pointing faces of these three proteins are mostly positively charged (Supplementary Fig. 4), which might be needed for these minor capsid proteins to make contacts with the negatively charged outer surface of the inner viral membrane.

**Viral assembly**. The viral capsid appears to be assembled with the help of the minor capsid proteins. Three rows of P3 and three copies of P2 construct a scaffold that glues together all the capsomers in each trisymmetron (Fig. 1b, c). The N terminus of each P2 protein extends to a neighboring pentasymmetron and glues together one-fifth of the pseudo-hexameric capsomers in a pentasymmetron (Fig. 1c). Neighboring symmetrons are bound together by the zip proteins (P11) (Fig. 1c). Other minor capsid proteins are recruited to the viral capsid either to stabilize the viral capsid further, to associate the inner viral membrane with the outer capsid shell, or to be involved in both (Fig. 1c and Supplementary Table 3). P2 and the zip protein (P11) play crucial roles in the whole process. P2 essentially acts as a molecular tape measure (or ruler) and determines the size of the viral capsid. The zip protein binds neighboring symmetrons together. Previous cryo-EM studies[5,6] showed that PBCV-1 has a unique vertex. To account for how the viral capsid is assembled around the unique vertex, it will be necessary to make a high-resolution 5-fold averaged reconstruction of the viral capsid. However, to achieve this, twelve times more data will be required.

Previous studies showed that bacteriophage PRD1[41] and Bam35[42] also use fiber-like tape-measure proteins to determine the size of the viral capsid. Such tape-measure proteins were first identified and modeled while interpreting the 4.0 Å X-ray structure of the bacteriophage PRD1 capsid. However, the identity of the Bam35 tape-measure protein is uncertain because of the limited resolution (7.3 Å) of the cryo-EM map. Like the tape-measure protein P2 of PBCV-1, the tape-measure proteins of both PRD1 and Bam35 adopt extremely extended fiber-like conformations, running roughly along each edge of the icosahedral faces, binding neighboring capsomers together. The major difference between the tape-measure protein P30 of PRD1 and P2 of PBCV-1 is that P30 of PRD1 forms a dimer through its

N-terminal 30 residues, whereas P2 of PBCV1 is a monomer. In addition, the PBCV-1 P2 protein is much longer than the PRD1 P30 protein, which is consistent with the larger size of the PBCV-1 capsid.

## Discussion

PBCV-1 assembles its capsid by using the tape-measure protein (P2) to determine its capsid size and the zip protein (P11) to bind neighboring symmetrons together. Most likely other approximately icosahedral NCLDVs have a similar assembly pathway and hence also contain a homologous tape-measure protein and a homologous zip protein. Like the zip protein (P11) of PBCV-1, a previous cryo-EM study showed that the zip protein of CIV also locates under the boundaries between neighboring symmetrons[7], presumably for gluing neighboring symmetrons together. BLAST searches[43] identified homologs of the tape-measure protein (P2) and zip protein (P11) of PBCV-1 in viruses belonging to Mimiviridae, such as APMV and CroV. Previous cryo-EM studies showed that the diameters of APMV[11] and CroV[10] are ~500 and ~300 nm, respectively. The viral capsid of PBCV-1 has a diameter of ~190 nm. The portion of the tape-measure protein of PBCV-1 that is involved in making contacts with the major capsid shell has ~473 amino acids (residues 94–566). Assuming the tape-measure protein homologs from APMV and CroV also exhibit extremely extended conformations as does the tape-measure protein of PBCV-1, we conclude the polypeptide chains of the tape-measure protein of APMV and CroV should contain ~1250 and ~750 amino acids, respectively. The polypeptide chains of the tape-measure protein of APMV (gene product of *L454*) and CroV (gene product of *crov185*) contain 1257 and 869 amino acids, respectively, close to their estimated lengths.

The icosahedrally averaged near-atomic structure of PBCV-1 shows that the PBCV-1 capsid is stabilized by a dense array of minor capsid proteins (P2–P14) inside the major capsid shell. This minor capsid protein shell cements most of the gaps between neighboring capsomers and uses transmembrane helices to mediate inner viral membrane association. The tape-measure protein P2 and the zip protein P11 play crucial roles in determining the capsid size and binding neighboring symmetrons together, respectively. Homologs of the tape-measure protein and the zip protein were also found in viruses belonging to Mimiviridae. Thus, other approximately icosahedral NCLDVs are very likely to have similar capsid assembly pathways. The use of tape-measure proteins to determine viral capsid size has also been observed in other viruses such as bacteriophage PRD1[41] and Bam35[42]. Thus, tape-measure proteins are probably used by many other large dsDNA viruses for capsid size determination.

## Methods

**Virus sample preparation**. The production and purification of virus PBCV-1 were described elsewhere[44]. Briefly, PBCV-1 virus was grown in *Chlorella variabilis* NC64A cells (**ATCC** 50258), and then purified by 1% Triton X-100 treatment and two successive rounds of gradient centrifugation in 10–40% sucrose. The virus was treated with proteinase K (0.02 mg/ml) between sucrose gradients to remove external contaminating proteins. After purification, the virus was re-suspended in virus suspension buffer (50 mM Tris-HCl, 10 mM MgCl$_2$, pH 7.8), filter sterilized (0.45 μm filter) and stored at 4 °C. Virus concentrations were determined by plaque assays (PFU/ml) and using a UV spectrophotometer as A$_{260}$ values.

**Cryo-EM data collection and processing**. The PBCV-1 sample was frozen onto Lacey carbon EM grids using a Gatan CP3 freezer with a blotting time of 6 s. The frozen grids were loaded into an FEI Titan Krios EM operated at 300 kV equipped with a Gatan K2 Summit detector. Automatic data collection using the Leginon program[45] and the Appion program[46] was performed using a magnification of 18,000x in the super-resolution mode, which resulted in a pixel size of 0.81 Å. The dose rate was ~8 e-/(pixel·s). A total of 5624 movies, each composed of 40 frames, were collected. Each frame has an exposure time of 200 ms. The movies were

subjected to motion correction using the MotionCorr program[47]. Micrographs were produced by summing up the aligned and dose-weighted frames of each movie. The contrast transfer function (CTF) parameters of each micrograph were estimated using the program CTFFIND3[48]. A total of 13,807 particles were picked using the Appion program[46]. The picked particles were subjected to reference-free two-dimensional (2D) classification using the RELION program[49]. After 2D classification, around 13,000 particles were selected for further processing. A previously reported icosahedrally averaged cryo-EM reconstruction of PBCV-1[6], low pass filtered to 40 Å resolution, was used as an initial model for determining the orientation and center of each particle and further refined using the jspr program[25], giving a resolution of 4.4 Å. To account for the defocus gradient and potential flexibility of the virus particles, a program[24] was designed to refine the structure further, resulting in a final resolution of 3.5 Å. The resolution was estimated by post process procedure in the RELION program[49] after applying a soft mask around the capsid shell.

**Model building and refinement**. The cryo-EM density representing the MCPs was interpreted by fitting a previously reported crystal structure of the MCP, Vp54, into the density map[21]. Because the densities of the glycans of the MCPs are poor, no attempt was made to model the glycans. The fitted atomic models were then rebuilt manually in COOT[26]. Cα models of the penton protein and the minor capsid proteins were built using EMbuilder[28] and rebuilt manually with COOT[26]. The Cα models that contain more than 60 amino acids and have prominent side-chain information in the density map were subjected to searching for satisfactory amino acid sequence fits using a Python script (see below) (Supplementary Fig. 3b). All the atomic models were refined using real-space refinement in PHENIX[27] (Supplementary Table 4).

**Alignment of the cryo-EM map density with protein sequences**. A Python script was written to compare the visual description of side chains as seen in the cryo-EM map with potential amino acid sequences identified by a proteomic study of the PBCV-1 virion[19]. The residues, which have good side-chain density, were reduced to numbers from 0 to 6 that described the size of the side chains as estimated by eye. The regions of poor density were classified into three different situations and treated differently. (1) When a region did not have good side-chain density, but had reliable Cα positions, the program was told that there was an amino acid here that might be any of the 20 possibilities. (2) When a region had good main-chain connectivity, but did not have reliable Cα atom placement, Cα atoms were tentatively built into the density. The total number of placed Cα atoms was taken as an estimate of how many amino acids there were in this region. Assuming the estimated number was $n$, the program would be told there were between $n − x/2$ and $n + x$ residues in this region, where $x$ was the maximum possible error. (3) When the density in a region was completely missing or so poor that the user had no idea where to place the Cα atoms, only the density before or after this region was used. The known protein sequences were similarly reduced to numbers (from 0 to 6) indicative of the size of each amino acid. The script was then used to search the side-chain size distribution of the polypeptide against that of the known protein sequences. Each aligned residue was scored according to Supplementary Table 5. The final score of each trial alignment was calculated by averaging the scores of all the aligned residues. A trial alignment was excluded from the list of possible alignments when encountering the following situations: (1) The potential amino acid sequence was shorter than the polypeptide chain seen in the cryo-EM map; (2) The side-chain size number of any residue as estimated using the cryo-EM map was larger than the side-chain size number of the aligned residue of the potential amino acid sequence by 2 or more. When two or more different regions of the cryo-EM map appeared to represent the same sequence (multiple sites for the same polypeptide within each icosahedral asymmetric unit), the density information of these polypeptide chains was combined to search for the target sequence when satisfying the following conditions: (1) these polypeptide chains had similar main-chain conformations; (2) the density information of each of these polypeptide chains had their best match with the same sequence candidate. However, in the case of the minor capsid protein P11, the ordered parts of its 12 copies were very short. Therefore, in this case, the density information of these polypeptide chains was grouped together based on the similarity of their main-chain conformations and on how they bound to their neighboring MCPs.

Two situations occurred in making the final sequence assignment of the identified minor capsid proteins and the penton protein: (1) there was only one sequence candidate that had possible alignments with the density information; (2) more than one protein sequence candidate had possible alignments with the density information. In the second case, a sequence was assigned based on two conditions: (1) the best alignment score of this sequence was larger than that of any other possible alignments of other sequence candidates by more than 0.1; (2) all the possible sequence assignments of other sequence candidates could be rejected using protein secondary structure prediction.

The Python script was tested on a few randomly chosen MCP cryo-EM densities before being applied to other proteins with unknown identity. Since information on the local chemical environment of any trial alignment was not used by the script, large inter- or intramolecular hydrophobic interfaces observed for any preferred alignment were used to validate the sequence found by the program.

Secondary structure predictions were also used to support the peptide identity. Furthermore, the structures of some of the identified proteins were found to have potential inter- or intramolecular disulfide bonds (Supplementary Table 2). Among the 14 proteins identified using the Python script mentioned above, five proteins were identified with potential intramolecular disulfide bonds and one with an intermolecular disulfide bond (Supplementary Table 2). The existence of some of the disulfide bonds predicted by the Python script was validated by mass spectrometry (Supplementary Fig. 5 and Supplementary Table 2). Two intramolecular disulfide bonds were identified for the P7 protein, C120-C134 and C188-C211. The product-ion spectra (MS/MS) for each of these peptides (Supplementary Fig. 5a, b) confirm the disulfide bonds present in this protein. Similarly, the loop-linked peptide containing C194-C201 was identified, and the product-ion spectra suggest the presence of a disulfide bond between the two cysteines (Supplementary Fig. 5c). Product-ion spectra from the peptide containing the inter-molecular disulfide bond (C215-C215) (Supplementary Fig. 5d) further confirmed the presence of the P10 protein in the cryo-EM map.

**Mass spectrometry**. Verification of the disulfide linkages was by mass spectrometry. Free cysteines were made unreactive by derivatizing with *N*-ethylmaleimide (NEM), and peptides with S−S bonds were then identified by a bottom−up proteomics strategy.

An aliquot (100 µL) of PBCV-1 virus in 50 mM HEPES, 10 mM MgCl₂, pH 7.8 was diluted with 20 µL of 50 mM HEPES buffer to a total volume of 120 µL and incubated at 25 °C for 1 h with shaking at 350 rpm in a thermomixer. After an aliquot (6 µL) of 1 M Tris-HCl buffer was added, the sample was heated at 60 °C for 60 min with shaking at 500 rpm and then spun down at 10,000×g for 10 min. The supernatant was taken and transferred to a new tube. Proteins were precipitated with excess acetone and stored at −20 °C until the NEM reaction and protease digestion. The sample preparation and NEM reaction were performed following a previously reported protocol[50]. The thawed sample was vortexed for a few sec, centrifuged at 15,000×g for 30 min, the acetone decanted, and the tube air-dried at room temperature for ~15 min. The sample was resuspended in 15 µL of 8 M urea in 100 mM Tris buffer, pH 7.5 to which was added NEM to 4 mM, incubated at 37 °C for 2 h. After the NEM reaction, the sample was removed from the incubator, centrifuged briefly, adjusted to 2 M urea concentration with 100 mM Tris buffer, pH 7.5, containing 2.5 mM NEM, and sequentially digested with Lys-C/Trypsin mix followed by Glu-C proteases. First, the Lys-C/trypsin mix (Promega) was added (1:20 protease: protein w/w) and incubated at 37 °C for 12 h followed by the addition of Glu-C protease (1:20 w/w) and incubated at 37 °C for 10 h.

An aliquot (5 µL) of the digested proteins was diluted 10-fold in Solvent A (water with 0.1% formic acid) and ~0.5 µg of the sample was loaded onto the column for LC-MS/MS analysis. The sample was analyzed three times with a Q Exactive plus mass spectrometer (Thermo Fisher Scientific) attached in line with an Ultimate 3000 Nano LC system (Thermo Scientific Dionex). Peptides were trapped and desalted in a guard column (Acclaim PepMap100, 100 µm × 2 cm, C18, 5 µm, 100 Å; Thermo Scientific Dionex) in Solvent A and were separated on a custom-packed C18 reversed-phase column (ProntoSIL aq, 0.075 mm × 150 mm, 3 µm, 120 Å, BISCHOFF chromatography, Germany). Peptides were separated over a linear 98-min gradient from 2–90% solvent B (80% acetonitrile, 20% water, 0.1% formic acid) and sprayed into the mass spectrometer with a spray voltage of 3.1 kV. The instrument was operated in positive-ion mode with a scan range for ions of $m/z$ 400–2000. Full mass spectra were acquired ($R = 70,000$ at $m/z$ 200), with automatic gain control set at $5 \times 10^5$ ions and a maximal injection time of 200 ms. Data-dependent product-ion HCD spectra were collected ($R = 17,500$ for $m/z$ 200) for the 10 most abundant precursor ions by using an isolation window of 2.0 $m/z$ and a normalized collision energy of 30%.

**NEM data analysis and search parameters**. Data were searched with pLink 2.0 software[50,51] against the PBCV-1 database containing 148 proteins with default pLink settings with few changes. Briefly, flow type for identification was set to disulfide bond (HCD-SS) with SS as the cross linker. Glu-C and trypsin were chosen as the enzyme, and the number of missed cleavages to 5. The peptide length was set between 4 and 30 with peptide mass set between 400 and 3000 Da. Precursor and fragment mass tolerances were set to ±20 ppm, respectively, and *N*-ethylmaleimide of cysteine, *N*-ethylmaleimide + water of cysteine. Oxidation of methionine, deamidation of asparagine and glutamine were included as variable modifications. The searched data were further filtered at a mass tolerance of ±10 ppm and false discovery rate of ≤5% at the spectra level. Only peptides identified with multiple product-ion (MS/MS) spectra were included in the result.

**Code availability**. All scripts and programs developed for this work are available at https://github.com/fgyfql/seqFinder

**Reporting Summary**. Further information on experimental design is available in the Nature Research Reporting Summary linked to this article.

## Data availability

The atomic coordinates for the major and minor capsid proteins and the cryo-EM density map have been submitted to the Protein Data Bank and Electron Microscopy Data Bank with accession codes 6NCL and EMD-0436, respectively. The authors declare that all other data supporting the findings of this study are available within the article and its Supplementary Information files or are available from the authors upon request.

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

## Acknowledgements

We thank Zhiqing Wang, Feng Long, S. Saif Hasan, and Andrei Fokine for helpful discussions and Sheryl Kelly for help in the preparation of this manuscript. This work was supported by National Institutes of Health (NIH) grant AI011219 to M.G.R., the National Key R&D Program of China (2017YFA0504700) to X.Z., and the Strategic Priority Research Program of CAS (XDB08030204) to X.Z. The mass spectrometry was also supported by NIH grant P41 GM103422 to M.L.G.

## Author contributions

M.G.R. conceived the project; X.Z. conceived essential techniques for obtaining a near-atomic resolution cryo-EM map; J.L.V.E. and I.A. provided the PBCV-1 sample; Q.F., T. K., Z.C. and Y.S. collected the cryo-EM data; Q.F., D.Z., and X.Z. processed the cryo-EM data; Q.F. interpreted the cryo-EM map; Y.L. and T.K. provided suggestions on cryo-EM data processing, cryo-EM map interpretation, and manuscript preparation; M.L.G. and J. A. performed the mass spectrometry analysis; M.G.R. and Q.F. wrote the initial draft with editing by J.L.V.E. and M.L.G.

## Additional information

**Competing interests:** The authors declare no competing interests.

