## [Peer Review File · Nature Communications]

Reviewers' Comments:

Reviewer #1:

Remarks to the Author:

This manuscript presents cryo-EM studies of a giant virus *Paramecium bursaria chlorella virus 1* (PBCV-1), which is the first cryo-EM structure of a nucleocytoplasmic large DNA virus (NCLDV) at near-atomic resolution. NCLDV infect many eukaryotic hosts, but their cryo-EM structures are hard to obtain at high resolution (better than 5Å), mostly due to their large size (up to 5,000 Å in diameter) and flexibility. Now, in this manuscript, Fang et al obtained the cryo-EM map of a relatively “small” member of the NCLDV family, PBCV-1, at 3.5 Å resolution. They were also able to build atomic models for the major capsid protein (MCP) and pentons (P1) assisted by some prior X-ray crystallographic structures. In addition to that, using self-developed algorithms the authors were also able to identify 13 different minor capsid proteins beneath the MCPs. The structural studies appear to be well designed and executed and reflect the long history that this lab has had in determining viral structures. Given the novelty of the present structure, I feel that it will be of general interest. Below are some comments and suggestions that need to be addressed in a revised manuscript.

Major point:

(1) My biggest concern is the model quality of the minor capsid proteins. First, the authors didn't mention what has been included in the deposited atomic coordinates. Were there only MCPs and P1, or were minor capsid proteins also included? Second, the authors only described that the map was estimated to be 3.5 Å resolution based on half-maps agreement. But the authors never mentioned what mask has been used for this resolution estimation. The resolution for different regions of the virus will be different and the authors need to discuss this. Third, the authors showed the 3D model of P1 in Fig. 1a and it looked reasonable. But the 3D models of P2-P14 seemed to contain much less secondary structure. For example, in Fig. 4, it seems there are many loops in proteins P6-P10. The authors need to present more data to convince readers that the minor capsid proteins were not mistraced. To better present the data, the following structural and refinement statistics should be included in the paper: (1) the map:model FSC for each individual minor capsid protein; (2) The real-space correlation coefficient for each minor capsid protein against the map filtered to the determined resolution; (3) the density for P10 with the model built into it to show that at the present resolution the trace is unambiguous.

Minor points

(2) In the abstract, the authors stated: “The structure consists of ~5040 copies of the major capsid protein, ~60 copies of the penton protein and ~1800 minor capsid proteins of which there are 13 different types.” Why are there “~” symbols?

(3) In Table S4, the authors state that the Mag is 18,000 and the pixel size is 0.81 Å/px. The magnification cannot be true. The authors mention in the table they collected the data under super-resolution mode, so they also need to address whether the pixel size stated is the physical pixel size or not.

(4) The MolProbity score and the clash-score in Table S4 are very high for this resolution (3.5Å), suggesting that the atomic models were poorly refined. Apparently the model needs to be refined further to clear these geometry problems.

Reviewer #2:

Remarks to the Author:

The authors have determined the structure of the giant PBCV1 capsid from purified virions using cryo-EM at 3.5 Å resolution. This break-through was made possible by using a recently published block-based reconstruction that allows correcting for the defocus gradient and flexibility in these large capsids. The complexity of the capsid and detailed analysis of all of the minor capsid proteins method is impressive. The identities of the minor proteins were established by first visually examining the sizes of the side chains and then matching the size on scale 0-6 to peptides derived by MS. This is a neat idea and some results were validated by MS of the disulphide bonds. The main finding of the study is how the minor proteins stabilize the 20 faces of the icosahedral capsid in this giant virus and how the size of these faces is limited by tape-measure proteins, analogous to what was originally discovered in PRD1.

Major comments:

The authors should comment on the error margins in their custom Python script based sequence assignment analysis. Best would be to implement an E-score similar to BLAST – i.e. what is the likelihood that a certain assignment score or better could be derived by pure chance in the set of sequences used? Showing the second highest score is not entirely satisfactory. How were loops / areas missing/unclear in the density treated. Also, this assignment should be explained better in the light of MS results. It is not clear if all of the proteins were identified by MS and if not, which ones were.

It is a bit disappointing that the authors haven't calculated a C5 reconstruction from their data to show the unique vertex. They simply state that 12 times more data would be required – but surely with the data they already have the previous structure of the unique vertex could be improved. This would increase the impact of the paper beyond what is currently presented. It is not however required to support the current findings, except for the copy numbers of the proteins. Would some proteins be in lesser amounts in the capsid if the special vertex is taken into account? Where proteins making the unique vertex identified by MS?

For this journal, it would make sense to divide the text in Introduction, Results and Discussion sections. Discussion would benefit from a conclusion paragraph. Now the manuscript finishes abruptly.

Minor comments.

L 42 (and elsewhere): Why is "icosahedral" in quotations? Perhaps because the word is not accurate here? It would be more accurate to say "NCLDVs with icosahedrally-symmetric capsids".

L 43–45: ICTV recommends that virus names when referring to physical entities are spelled in lower case (e.g. faustovirus) unless they contain a proper noun (e.g. Melbourne virus). See <https://talk.ictvonline.org/information/w/faq/386/how-to-write-a-virus-name>

L 65–66: "Various assembly pathways". Perhaps the authors could put forward a more specific hypothesis. For example "how is the size of the icosahedrally-symmetric capsid controlled during assembly".

L 80: The T number is not a "quasi-symmetry", rather it defines the type of triangulation the (quasi-equivalent) capsomers follow. So better to say "arrangement of the capsomers in the icosahedral lattice follows the T=169d triangulation" and refer to Caspar and Klug.

L 80: Consider moving the values of the two icosahedral lattice indices (h and k) to a figure legend

and show them visually in the figure – they are not clear to a general reader.

L 84–86 and L98–99: Makes and models of microscopes and detectors seem strange in the Introduction and Results. Move these to Methods and simply state “collecting data using a direct electron detector”.

L 105: Typo (icosahedral)

L 141 and elsewhere: Replace incorrect “electron density” with “cryo-EM density map” or just “density”. (Cryo-EM maps are not just “electron density” as the beam electrons interact with both sample atom electrons and nuclei.)

L267: It would be interesting to see if the TM regions become visible if the map is locally filtered to lower resolution using for instance Relion’s or Xmipp’s local resolution / filtering tools.

L290: Tape measure protein has also been detected in Bam35 phage and possibly other viruses. It would be useful to extend the comparisons to these as tape-measure protein is a very interesting observation and one of the main results.

Response to Reviewers

NCOMMS-18-26039-T

Your manuscript entitled "First near-atomic structure of a giant virus" has now been seen by 2 referees, whose comments are appended below. You will see from their comments copied below that while they find your work of considerable potential interest, they have raised quite substantial concerns that must be addressed. In light of these comments, we cannot accept the manuscript for publication, but would be interested in considering a revised version that addresses these serious concerns.

We are somewhat puzzled by your comments above. As we read the comments of the reviewers they appear to us to be rather positive. The changes they suggest are fairly minor although they required about two weeks of work for iterative improvement of the structural model and to perform a 5-fold reconstruction in addition to the existing icosahedral reconstruction presented by us here. No matter whether these are “major” or “minor” changes, we have made these changes as described below.

In particular, a revised manuscript would need to provide additional analyses to improve the model quality of the minor capsid proteins and sequence alignment analysis, along the lines suggested by reviewers. Please also address concerns on copy numbers of proteins and improve refinement of atomic models. Please make sure that a revised manuscript addresses these and all other concerns in full.

We have now further refined and significantly improved the structures of all the minor and major capsid proteins, calculated a five-fold reconstruction of the viral capsid, added more details of the sequence assignment as explained below.

Reviewers' comments:

Reviewer #1 (Remarks to the Author):

*This manuscript presents cryo-EM studies of a giant virus *Paramecium bursaria chlorella virus 1* (PBCV-1), which is the first cryo-EM structure of a nucleocytoplasmic large DNA virus (NCLDV) at near-atomic resolution. NCLDV infect many eukaryotic hosts, but their cryo-EM structures are hard to obtain at high resolution (better than 5Å), mostly due to their large size (up to 5,000 Å in diameter) and flexibility. Now, in this manuscript, Fang et al obtained the cryo-EM map of a relatively “small” member of the NCLDV family, PBCV-1, at 3.5 Å resolution. They were also able to build atomic models for the major capsid protein (MCP) and pentons (P1) assisted by some prior X-ray crystallographic structures. In addition to that, using self-developed algorithms the authors were also able to identify 13 different minor capsid proteins beneath the MCPs. The structural studies appear to be well designed and executed and reflect the long history that this lab has had in determining viral structures. Given the novelty of the present structure, I feel that it will be of general interest. Below are some comments and suggestions that need to be addressed in a revised manuscript.*

Thank you

Major point:

(1) My biggest concern is the model quality of the minor capsid proteins. First, the authors didn't mention what has been included in the deposited atomic coordinates. Were there only MCPs and P1, or were minor capsid proteins also included?

Models of the MCPs, P1 (the penton protein) as well as the P2 to P14 minor capsid proteins had been included in the atomic coordinates deposited with the PDB. This information has now been clearly stated in the PDB data deposition statement (see lines 491-493).

Second, the authors only described that the map was estimated to be 3.5 Å resolution based on half-maps agreement. But the authors never mentioned what mask has been used for this resolution estimation.

This resolution was estimated by applying a soft mask around the viral capsid shell. This information has now been added in the methods section (see lines 366-367).

The resolution for different regions of the virus will be different and the authors need to discuss this.

The local resolution within one asymmetric unit of the icosahedrally averaged cryo-EM map has now been shown in Supplementary Fig. 1b.

Third, the authors showed the 3D model of P1 in Fig. 1a and it looked reasonable. But the 3D models of P2-P14 seemed to contain much less secondary structure. For example, in Fig. 4, it seems there are many loops in proteins P6-P10.

A common property of the minor capsid proteins in PBCV1 and in other large dsDNA viruses is the occurrence of loops and extended polypeptide chains. Most of these loops fill gaps between neighboring capsomers and are stabilized by making contacts with neighboring major capsid proteins. Similar characteristics are also possessed by the tape-measure protein P30 of bacteriophage PRD1 (see ref# 41).

The authors need to present more data to convince readers that the minor capsid proteins were not mistraced. To better present the data, the following structural and refinement statistics should be included in the paper:

(1) the map:model FSC for each individual minor capsid protein;

The FSC for each minor capsid protein has now been calculated as a function of resolution. The resolutions are shown in the Supplementary Table 1 for FSCs of 0.25 and 0.50.

(2) The real-space correlation coefficient for each minor capsid protein against the map filtered to the determined resolution;

This has now been calculated and added to Supplementary Table 1 assuming an overall resolution of 3.5 Å.

(3) the density for P10 with the model built into it to show that at the present resolution the trace is unambiguous.

This has now been shown in Supplementary Fig. 20 for each of the two conformations of this structure.

Minor points

(2) In the abstract, the authors stated: “The structure consists of ~5040 copies of the major capsid protein, ~60 copies of the penton protein and ~1800 minor capsid proteins of which there are 13 different types.” Why are there “~” symbols?

These copy numbers are not accurate if the unique vertex of the virion were taken into account. However, because we are considering here the icosahedrally averaged structure these “~” symbols have now been removed and therefore the numbers refer to a hypothetical exactly icosahedral structure.

(3) In Table S4, the authors state that the Mag is 18,000 and the pixel size is 0.81 Å/px. The magnification cannot be true. The authors mention in the table they collected the data under super-resolution mode, so they also need to address whether the pixel size stated is the physical pixel size or not.

We did use a Mag of 18,000 when collecting the data. The physical pixel size was 1.62 Å/px, resulting in a super-resolution pixel size of 0.81Å/px. This has now been addressed in Supplementary Table 4.

(4) The MolProbity score and the clash-score in Table S4 are very high for this resolution (3.5Å), suggesting that the atomic models were poorly refined. Apparently the model needs to be refined further to clear these geometry problems.

The model has now been refined further and significantly improved. The current MolProbity score and clash score are 1.71 and 10.16, respectively. These statistics has now been updated in Supplementary Table 4.

Reviewer #2 (Remarks to the Author):

The authors have determined the structure of the giant PBCV1 capsid from purified virions using cryo-EM at 3.5 Å resolution. This break-through was made possible by using a recently published block-based reconstruction that allows correcting for the defocus gradient and flexibility in these large capsids. The complexity of the capsid and detailed analysis of all of the minor capsid proteins method is impressive. The identities of the minor proteins were established by first visually examining the sizes of the side chains and then matching the size on scale 0-6 to peptides derived by MS. This a neat idea and some results were validated by MS of the disulphide bonds. The main finding of the study is how the minor proteins stabilize the 20 faces of the icosahedral capsid in this giant virus and how the size

of these faces is limited by tape-measure proteins, analogous to what was originally discovered in PRD1.

Thank you

Major comments:

The authors should comment on the error margins in their custom Python script based sequence assignment analysis. Best would be to implement an E-score similar to BLAST – i.e. what is the likelihood that a certain assignment score or better could be derived by pure chance in the set of sequences used? Showing the second highest score is not entirely satisfactory.

Finding the sequence of each polypeptide chains was based on alignment score as well as an exclusion principle. The second highest score was just one condition in this exclusion process. Thus, it is not obvious how to calculate an overall probability for any given alignment. We have now added more details to make this clear (see lines 399 – 403, 413 – 420). Successful sequence assignments depended on the number and quality of side-chain densities. For some polypeptide chains whose ordered parts were very short or overall density quality was very poor, we were not able to interpret them or find their sequences (This was mentioned in lines 374 – 376. These uninterpreted densities are now shown in Supplementary Fig. 3b).

How were loops / areas missing/unclear in the density treated.

The details on how such areas were treated have now been added in the manuscript. (see lines 385 – 394)

Also, this assignment should be explained better in the light of MS results.

(1) Initially, we tried to use cross-linking MS to do this. The idea is to treat the virus particles with cross-linking reagents followed by protease digestion. MS is then used to detect intermolecularly cross-linked peptides to give some clues about the architecture of the virion. However, although we have been working on this for more than one year, we have not had any success. This is probably because the minor capsid proteins are inside the virion. Thus, the crosslinking reagents were not able to penetrate into the virion. This should be a general problem for identifying the minor capsid proteins of these large dsDNA viruses using MS. We will be grateful if the reviewer can suggest any better ways.

(2) After the minor capsid proteins were identified by using the Python script, we found that there were adjacent cysteine residues which could form potential disulfide bonds in some of the models. Three out of the fourteen identified proteins were validated via detecting the existence of disulfide bonds by MS. This showed that the procedure we used to identify the minor capsid proteins was reasonable.

(3) The sequence assignments were supported by secondary structure predictions and validated by intra- and intermolecular chemical environments. Such information was not used by the python script when searching for the target sequences.

It is not clear if all of the proteins were identified by MS and if not, which ones were.

Only three out of the fourteen sequence assignments were validated by MS. This information had been shown in Supplementary Table 2. Other validation procedures were used for the rest of the proteins as described in lines 422 – 426.

It is a bit disappointing that the author's haven't calculated a C5 reconstruction from their data to show the unique vertex. They simply state that 12 times more data would be required – but surely with the data they already have the previous structure of the unique vertex could be improved. This would increase the impact of the paper beyond what is currently presented. It is not however required to support the current findings, expect for the copy numbers of the proteins. Would some proteins be in lesser amounts in the capsid if the special vertex is taken into account?

A five-fold averaged cryo-EM map has been calculated. But the resolution is only 8.7 Å. We will attempt to push the resolution of the five-fold reconstruction in our future work. This require about 12 times the amount of data used in the present analysis.

Where proteins making the unique vertex identified by MS?

Considering the resolution of the 5-fold averaged map is only 8.7 Å, identifying a specific protein by MS is probably an impossible task at this time. However, we will attempt to identify the specific proteins at the unique vertex in our future work.

For this journal, it would make sense to divide the text in in Introduction, Results and Discussion sections.

This has now been done.

Discussion would benefit from a conclusion paragraph. Now the manuscript finishes abruptly.

This has now been added (see lines 326 – 336).

Minor comments.

L 42 (and elsewhere): Why is “icosahedral” in quotations? Perhaps because the word is not accurate here? It would be more accurate to say “NCLDV with icosahedrally-symmetric capsids”.

We use quotation marks to show that the icosahedral symmetry is only approximate in view of the special vertex.

L 43–45: ICTV recommends that virus names when referring to physical entities are spelled in lower case (e.g. faustovirus) unless they contain a proper noun (e.g. Melbourne virus). See <https://talk.ictvonline.org/information/w/faq/386/how-to-write-a-virus-name>

Thank you very much for the information. This has been corrected.

L 65–66: “Various assembly pathways”. Perhaps the authors could put forward a more specific hypothesis. For example “how is the size of the icosahedrally-symmetric capsid controlled during assembly”.

This has been done. (See lines 67 – 68)

L 80: The T number is not a “quasi-symmetry”, rather it defines the type of triangulation the (quasi-equivalent) capsomers follow. So better to say “arrangement of the capsomers in the icosahedral lattice follows the $T=169d$ triangulation” and refer to Caspar and Klug.

Thank you very much for this comment. This has now been modified as suggested. (see lines 81 – 82)

L 80: Consider moving the values of the two icosahedral lattice indices (h and k) to a figure legend and show them visually in the figure – they are not clear to a general reader.

This has now been shown in Figure 1a.

L 84–86 and L98–99: Makes and models of microscopes and detectors seem strange in the Introduction and Results. Move these to Methods and simply state “collecting data using a direct electron detector”.

This has been modified as suggested.

L 105: Typo (icosahedral)

This has been corrected.

L 141 and elsewhere: Replace incorrect “electron density” with “cryo-EM density map” or just “density”. (Cryo-EM maps are not just “electron density” as the beam electrons interact with both sample atom electrons and nuclei.)

This has been corrected.

L267: It would be interesting to see if the TM regions become visible if the map is locally filtered to lower resolution using for instance Relion’s or Xmipp’s local resolution / filtering tools.

We have tried this. No, these regions did not become visible.

L290: Tape measure protein has also been detected in Bam35 phage and possibly other viruses. It would be useful to extend the comparisons to these as tape-measure protein is a very interesting observation and one of the main results.

This extended comparison has now been added. (see lines 296 – 306) This point has also been mentioned in the discussion section. (see lines 333 – 336)

Reviewers' Comments:

Reviewer #1:

Remarks to the Author:

The authors have done a very good job in addressing my concerns and the revised paper is significantly improved.

Edward Egelman

Reviewer #2:

Remarks to the Author:

I am evaluating authors response and the modified manuscript. After the minor comment has been addressed in my view the manuscript would be suitable for publication.

Calculating the significance of the alignment score should have been relatively easy to add: for instance take the query 'sequence', randomise it and calculate the alignment score. Repeat 100 times. How much better is the real alignment vs. random alignments? If 100 random alignments have a lower score than the real alignment, the P-value is less than 0.01. It's a shame that the authors didn't follow up on this advice and some concern remains over the significance of the reported alignment scores (best and second best). However, the authors have now provided extensive detail what other factors were considered to select the best alignment. This is satisfactory albeit not very elegant.

Custom software was used in this study. It would be important to make sure that any custom software used is made available on GitHub or other repository.

Minor comment:

Figure 1A still requires a small modification. The indices and arrows should be mirrored so that $h=8$ and $k=7$ (by convention $h>0$, $k>1$, $h\geq k$). This way after taking 8 steps along axis H one needs to turn right (dextro) and then take 7 steps along axis K to find the next five-fold. Now the figure gives the wrong impression that the T number is laevo (as left turn is implicated). This confusion arises from the fact that in the figure h and k are the wrong way around ($h<k$).

REVIEWERS' COMMENTS:

Reviewer #1 (Remarks to the Author):

The authors have done a very good job in addressing my concerns and the revised paper is significantly improved.

Edward Egelman

Thank you.

Reviewer #2 (Remarks to the Author):

I am evaluated authors response and the modified manuscript. After the minor comment has been addressed in my view the manuscript would be suitable for publication.

Thank you. The minor comment has now been addressed.

Calculating the significance of the alignment score should have been relatively easy to add: for instance take the query 'sequence', randomise it and calculate the alignment score. Repeat 100 times. How much better is the real alignment vs. random alignments? If 100 random alignments have a lower score than the real alignment, the P-value is less than 0.01. It's a shame that the authors didn't follow up on this advice and some concern remains over the significance of the reported alignment scores (best and second best). However, the authors have now provided extensive detail what other factors were considered to select the best alignment. This is satisfactory albeit not very elegant.

Thank you.

Custom software was used in this study. It would be important to make sure that any custom software used is made available on GutHub or other repository.

This has now been done.

Minor comment:

Figure 1A still requires a small modification. The indices and arrows should be mirrored so that $h=8$ and $k=7$ (by convention $h>0$, $k>1$, $h>=k$). This way after taking 8 steps along axis H one needs to turn right (dextro) and then take 7 steps along axis K to find the next five-fold. Now the figure gives the wrong impression that the T number is laevo (as left turn is implicated). This confusion arises from the fact that in the figure h and k are the wrong way around ($h<k$).

This has now been corrected.